# Gradient based sample selection for online continual learning

**Rahaf Aljundi**[*]
KU Leuven
rahaf.aljundi@gmail.com

**Min Lin**
Mila
mavenlin@gmail.com

**Baptiste Goujaud**
Mila
baptiste.goujaud@gmail.com

**Yoshua Bengio**
Mila
yoshua.bengio@mila.quebec

## Abstract

A continual learning agent learns online with a non-stationary and never-ending stream of data. The key to such learning process is to overcome the catastrophic forgetting of previously seen data, which is a well known problem of neural networks. To prevent forgetting, a replay buffer is usually employed to store the previous data for the purpose of rehearsal. Previous works often depend on task boundary and i.i.d. assumptions to properly select samples for the replay buffer. In this work, we formulate sample selection as a constraint reduction problem based on the constrained optimization view of continual learning. The goal is to select a fixed subset of constraints that best approximate the feasible region defined by the original constraints. We show that it is equivalent to maximizing the diversity of samples in the replay buffer with parameters gradient as the feature. We further develop a greedy alternative that is cheap and efficient. The advantage of the proposed method is demonstrated by comparing to other alternatives under the continual learning setting. Further comparisons are made against state of the art methods that rely on task boundaries which show comparable or even better results for our method.

## 1 Introduction

The central problem of continual learning is to overcome the catastrophic forgetting problem of neural networks. Current continual learning methods can be categorized into three major families based on how the information of previous data are stored and used. We describe each of them below.

The *prior-focused* Prior-focused methods use a penalty term to regularize the parameters rather than a hard constraint. The parameter gradually drifts away from the feasible regions of previous tasks, especially when there is a long chain of tasks and when the tasks resemble each other [6]. It is often necessary to hybridize the prior-focused approach with the replay-based methods for better results [13, 6].

Another major family is the *parameter isolation* methods which dedicates different parameters for different tasks to prevent interference. Dynamic architectures that freeze/grow the network belong to this family. However, it is also possible to isolate parameters without changing the architecture [7, 11].

---

[*]Work mostly done while first author was a visiting researcher at Mila.

Both of the above strategies explicitly associate neurons with different tasks, with the consequence that task boundaries are mandatory during both training and testing. Due to the dependency on task boundaries during test, this family of methods tilts more towards multi-task learning than continual learning.

The *replay-based* approach stores the information in the example space either directly in a replay buffer or in a generative model. When learning new data, old examples are reproduced from the replay buffer or generative model, which is used for rehearsal/retraining or used as constraints for the current learning, yet the old examples could also be used to provide constraints [10].

Most of the works in the above three families use a relaxed task incremental assumption: the data are streamed one task at a time, with different distributions for each task, while keeping the independent and identically distributed (i.i.d.) assumption and performing offline training within each task. Consequently, they are not directly applicable to the more general setting where data are streamed online with neither i.i.d. assumption nor task boundary information. Both prior-focused methods and replay-based methods have the potential to be adapted to the general setting. However, we are mostly interested in the replay-based methods in this work, since it is shown that prior-focused methods lead to no improvement or marginal improvement when applied on top of a replay-based method. Specifically, we develop strategies to populate the replay buffer under the most general condition where no assumptions are made about the online data stream.

Our contributions are as follows: 1) We formulate replay buffer population as a constraint selection problem and formalize it as a solid angle minimization problem. 2) We propose a surrogate objective for it and empirically verify that the surrogate objective aligns with the goal of solid angle minimization 3) As a cheap alternative for large sample selection, we propose a greedy algorithm that is as efficient as reservoir sampling yet immune to imbalanced data stream. 4) We compare our method to different selection strategies and show the ability of our solutions to always select a subset of samples that best represents the previous history 5) We perform experiments on continual learning benchmarks and show that our method is on par with, or better than, the previous methods. Yet, requiring no i.i.d. assumptions or task boundaries.

## 2    Related Work

Our continual learning approach belongs to the replay based family. Methods in this family alleviate forgetting by replaying stored samples from previous history when learning new ones. Although storage of the original examples in memory for rehearsal dates back to 1990s [16], to date it is still a rule of thumb to overcome catastrophic forgetting in practical problems. For example, experience replay is widely used in reinforcement learning where the data distributions are usually non-stationary and prone to catastrophic forgetting [9, 12].

Recent works that use replay buffer for continual learning include iCaRL [14] and GEM [10], both of which allocate memory to store a core-set of examples from each task. These methods still require task boundaries in order to divide the storage resource evenly to each task. There are also a few previous works that deals with the situation where task boundary and i.i.d. assumption is not available. For example, reservoir sampling has been employed in [5, 8] so that the data distribution in the replay buffer follows the data distribution that has already been seen. The problem of reservoir sampling is that the minor modes in the distribution with small probability mass may fail to be represented in the replay buffer. As a remedy to this problem, coverage maximization is also proposed in [8]. It intends to keep diverse samples in the replay buffer using Euclidean distance as a difference measure.While the Euclidean distance may be enough for low dimensional data, it could be uninformative when the data lies in a structured manifold embedded in a high dimensional space. In contrast to previous works, we start from the constrained optimization formulation of continual learning, and show that the data selection for replay buffer is effectively a constraint reduction problem.

## 3    Continual Learning as Constrained Optimization

We consider the supervised learning problem with an online stream of data where one or a few pairs of examples $(x, y)$ are received at a time. The data stream is non-stationary with no assumption on the distribution such as the i.i.d. hypothesis. Our goal is to optimize the loss on the current example(s) without increasing the losses on the previously learned examples.

## 3.1 Problem Formulation

We formulate our goal as the following constrained optimization problem. Without loss of generality, we assume the examples are observed one at a time.

$$\theta^t = \underset{\theta}{\operatorname{argmin}} \, \ell(f(x_t; \theta), y_t) \tag{1}$$

$$\text{s.t. } \ell(f(x_i; \theta), y_i) \leq \ell(f(x_i; \theta^{t-1}), y_i); \; \forall i \in [0 \ldots t-1]$$

$f(.; \theta)$ is a model parameterized by $\theta$ and $\ell$ is the loss function. $t$ is the index of the current example and $i$ indexes the previous examples.

As suggested by [10], the original constraints can be rephrased to the constraints in the gradient space:

$$\langle g, g_i \rangle = \left\langle \frac{\partial \ell(f(x_t; \theta), y_t)}{\partial \theta}, \frac{\partial \ell(f(x_i; \theta), y_i)}{\partial \theta} \right\rangle \geq 0; \tag{2}$$

However, the number of constraints in the above optimization problem increases linearly with the number of previous examples. The required computation and storage resource for an exact solution of the above problem will increase indefinitely with time. It is thus more desirable to solve the above problem approximately with a fixed computation and storage budget. In practice, a *replay buffer* $\mathcal{M}$ limited to $M$ memory slots is often used to keep the previous examples. The constraints are thus only active for $(x_i, y_i) \in \mathcal{M}$. How to populate the replay buffer then becomes a crucial research problem.

Gradient episodic memory (GEM) assumes access to task boundaries and an i.i.d. distribution within each task episode. It divides the memory budget evenly among the tasks. i.e. $m = M/T$ slots is allocated for each task, where $T$ is the number of tasks. The last $m$ examples from each task are kept in the memory. This has clear limitations when the task boundaries are not available or when the i.i.d. assumption is not satisfied. In this work, we consider the problem of how to populate the replay buffer in a more general setting where the above assumptions are not available.

## 3.2 Sample Selection as Constraint Reduction

Motivated by Eq.1, we set our goal to selecting $M$ examples so that the feasible region formed by the corresponding reduced constraints is close to the feasible region of the original problem. We first convert the original constraints in 2 to the corresponding feasible region:

$$C = \bigcap_{i \in [0..t-1]} \{g | \langle g, g_i \rangle \geq 0\} \tag{3}$$

We assume here that $C$ is generally not empty. It is highly unlikely to happen if we consider a number of parameters much larger than the number of gradients $g_i$, except if we encounter an outlier that has different label $y_i$ with the same input $x_i$. In this work we don't consider the existence of outliers. Geometrically, $C$ is the intersection of the half spaces described by $\langle g, g_i \rangle \geq 0$, which forms a polyhedral convex cone. The relaxed feasible region corresponding to the replay buffer is:

$$\tilde{C} = \bigcap_{g_i \in \mathcal{M}} \{g | \langle g, g_i \rangle \geq 0\} \tag{4}$$

For best approximation of the original feasible region, we require $\tilde{C}$ to be as close to $C$ as possible. It is easy to see that $C \subset \tilde{C}$ because $\mathcal{M} \subset [g_0 \ldots g_{t-1}]$. We illustrate the relation between $C$ and $\tilde{C}$ in Figure 1.

On the left, $C$ is represented while the blue hyperplane on the right corresponds to a constraint that has been removed. Therefore, $\tilde{C}$ (on the right) is larger than $C$ for the inclusion partial order. As we want $\tilde{C}$ to be as close to $C$ as possible, we actually want the "smallest" $\tilde{C}$, where "small" here remains to be defined, as the inclusion order is not a complete order. A potential measure of the size of a convex cone is its solid angle defined as the intersection between the cone and the unit sphere.

$$\text{minimize}_{\mathcal{M}} \, \lambda_{d-1} \left( S_{d-1} \cap \bigcap_{g_i \in \mathcal{M}} \{g | \langle g, g_i \rangle \geq 0\} \right) \tag{5}$$

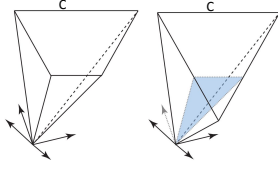

Figure 1: Feasible region (poly-hedral cone) before and after con-straint selection. The selected constraints (excluding the blue one) are chosen to best approxi-mate the original feasible region.

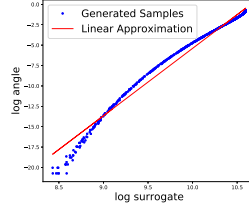

Figure 2: Correlation between solid angle and our proposed sur-rogate in 200 dimension space in log scale. Note that we only need monotocity for our objective to hold.

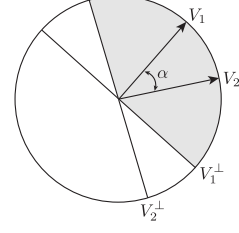

Figure 3: Relation be-tween angle formed by two vectors ($\alpha$) and the associ-ated feasible set (grey re-gion)

where $d$ denotes the dimension of the space, $S_{d-1}$ the unit sphere in this space, and $\lambda_{d-1}$ the Lebesgue measure in dimension $d - 1$.

Therefore, solving 5 would achieve our goal. Note that, in practice, the number of constraints and thus the number of gradients is usually smaller than the dimension of the gradient, which means that the feasible space can be seen as the Cartesian product between its own intersection with $span\,(\mathcal{M})$ and the orthogonal subspace of $span\,(\mathcal{M})$. That being said, we can actually reduce our interest to the size of the solid angle in the $M$-dimensional space $span\,(\mathcal{M})$, as in 6.

$$\text{minimize}_{\mathcal{M}} \; \lambda_{M-1} \left( S_{M-1}^{span(\mathcal{M})} \cap \bigcap_{g_i \in \mathcal{M}} \{g | \langle g, g_i \rangle \geq 0\} \right) \tag{6}$$

where $S_{M-1}^{span(\mathcal{M})}$ denotes the unit sphere in $span(\mathcal{M})$.

Note that even if the sub-spaces $span(\mathcal{M})$ are different from each other, they all have the same dimension as $M$, which is fixed, hence comparing their $\lambda_M$-measure makes sense. However, this objective is hard to minimize since the formula of the solid angle is complex, as shown in [15] and [2]. Therefore, we propose, in the next section, a surrogate to this objective that is easier to deal with.

### 3.3 An Empirical Surrogate to Feasible Region Minimization

Intuitively, to decrease the feasible set, one must increase the angles between each pair of gradients. Indeed, this is directly visible in 2D with Figure 3. Based on this observation, we propose the surrogate in Eq.9.

$$\text{minimize}_{\mathcal{M}} \sum_{i,j \in \mathcal{M}} \frac{\langle g_i, g_j \rangle}{\|g_i\| \|g_j\|} \tag{7}$$

$$s.t.\ \mathcal{M} \subset [0\,..\,t-1]; \; |\mathcal{M}| = M$$

We empirically studied the relationship between the solid angle and the surrogate function in higher dimensional space using randomly sampled vectors as the gradient. Given a set of sampled vectors, the surrogate value is computed analytically and the solid angle is estimated using Monte Carlo approximation of Eq.6. The results are presented in Figure 2, which shows a monotonic relation between the solid angle and our surrogate.

It is worth noting that minimization of Eq.9 is equivalent to maximization of the variance of the gradient direction as is shown in Eq.8.

$$\text{Var}_{\mathcal{M}} \left[ \frac{g}{\|g\|} \right] = \frac{1}{M} \sum_{k \in \mathcal{M}} \left\| \frac{g}{\|g\|} \right\|^2 - \left\| \frac{1}{M} \sum_{k \in \mathcal{M}} \frac{g}{\|g\|} \right\|^2 \tag{8}$$

$$= 1 - \frac{1}{M^2} \sum_{i,j \in \mathcal{M}} \frac{\langle g_i, g_j \rangle}{\|g_i\| \|g_j\|}$$

This brings up a new interpretation of the surrogate, which is maximizing the diversity of samples in the replay buffer using the parameter gradient as the feature. Intuitively, keeping diverse samples in the replay buffer could be an efficient way to use the memory budget. It is also possible to maximize the variance directly on the samples or on the hidden representations, we argue that the parameter gradient could be a better option given its root in Eq. 1. This is also verified with experiments.

### 3.4    Online Sample Selection

#### 3.4.1    Online sample selection with Integer Quadratic Programming.

We assume an infinite input stream of data where at each time a new sample(s) is received. From this stream we keep a fixed buffer of size $M$ to be used as a representative of the previous samples. To reduce computation burden, we use a "recent" buffer in which we store the incoming examples and once is full we perform selection on the union of the replay buffer and the "recent" buffer and replace the samples in the replay buffer with the selection. To perform the selection of $M$ samples, we solve Eq. 9 as an integer quadratic programming problem as shown in Appendix A.2. The exact procedures are described in algorithm 1. While this is reasonable in the cases of small buffer size, we observed a big overhead when utilizing larger buffers which is likely the case in practical scenario. The overhead, comes from both the need to get the gradient of each sample in the buffer and the recent buffer and from solving the quadratic problem that is polynomial w.r.t. the size of the buffer. Since this might limit the scalability of our approach, we suggest an alternative greedy method.

#### 3.4.2    An in-exact greedy alternative.

We propose an alternative greedy method based on heuristic, which could achieve the same goal of keeping diverse examples in the replay buffer, but is much cheaper than performing integer quadratic programming. The key idea is to maintain a score for each sample in the replay buffer. The score is computed by the maximal cosine similarity of the current sample with a fixed number of other random samples in the buffer. When there are two samples similar to each other in the buffer, their scores are more likely to be larger than the others. In the beginning when the buffer is not full, we add incoming samples along with their score to the replay buffer. Once the buffer is full, we randomly select samples from the replay buffer as the candidate to be replaced. We use the normalized score as the probability of this selection. The score of the candidate is then compared to the score of the new sample to determine whether the replacement should happen or not.

More formally, denote the score as $\mathcal{C}_i$ for sample $i$ in the buffer. Sample $i$ is selected as a candidate to be replaced with probability $P(i) = \mathcal{C}_i / \sum_j \mathcal{C}_j$. The replacement is a bernoulli event that happens with probability $\mathcal{C}_i / (c + \mathcal{C}_i)$ where $\mathcal{C}_i$ is the score of the candidate and $c$ is the score of the new data. We can apply the same procedure for each example when a batch of new data is received.

Algorithm 2 describes the main steps of our gradient based greedy sample selection procedure. It can be seen that the major cost of this selection procedure corresponds only to the estimation of the gradients of the selected candidates which is a big computational advantage over the other selection strategies.

### 3.5    Constraint vs Regularization

Projecting the gradient of the new sample(s) exactly into the feasible region is computationally very expensive especially when using a large buffer. A usual work around for constrained optimization is to convert the constraints to a soft regularization loss. In our case, this is equivalent to performing rehearsal on the buffer. Note that [4] suggests to constrain only with one random gradient direction from the buffer as a cheap alternative that works equally well to constraining with the gradients of the previous tasks, it was later shown by the same authors [5] that rehearsal on the buffer has a competitive performance. In our method, we do rehearsal while in Appendix B.2 we evaluate both rehearsal and constrained optimization on a small subset of disjoint MNIST and show comparable results.

### 3.6 Summary of the Proposed Approach

To recap, we start from the constrained optimization view of continual learning, which needs to be relaxed by constraint selection. Instead of random selection, we perform constraint selection by minimizing the solid angle formed by the constraints. We propose a surrogate for the solid angle objective, and show their relation numerically. We further propose a greedy alternative that is computationally more efficient. Finally, we test the effectiveness of the proposed approach on continual learning benchmarks in the following section.

---

**Algorithm 1** IQP Sample Selection

1: Input: $M_r$, $M_b$
2: **function** SELECTSAMPLES($\mathcal{M}$, $M$)
3:     $\hat{\mathcal{M}} \leftarrow \arg\min_{\hat{\mathcal{M}}} \sum_{i,j \in \hat{\mathcal{M}}} \frac{\langle g_i, g_j \rangle}{\|g_i\|\|g_j\|}$
4:     s.t. $\hat{\mathcal{M}} \subset \mathcal{M}$; $|\hat{\mathcal{M}}| = M$
5:     **return** $\hat{\mathcal{M}}$
6: **end function**
7: Initialize: $\mathcal{M}_r$, $\mathcal{M}_b$
8: Receive: $(x, y)$          ▷ one or few consecutive examples
9: Update($x, y, \mathcal{M}_b$)
10: $\mathcal{M}_r \leftarrow \mathcal{M}_r \cup \{(x, y)\}$
11: **if** len($\mathcal{M}_r$) $> M_r$ **then**
12:     $\mathcal{M}_b \leftarrow \mathcal{M}_b \cup \mathcal{M}_r$
13:     $\mathcal{M}_r \leftarrow \{\}$
14:     **if** len($\mathcal{M}_b$) $> M_b$ **then**
15:         $\mathcal{M}_b \leftarrow$ SelectSamples($\mathcal{M}_b, M_b$)
16:     **end if**
17: **end if**

---

**Algorithm 2** Greedy Sample Selection

1: Input: $n$, $M$
2: Initialize: $\mathcal{M}$, $\mathcal{C}$
3: Receive: $(x, y)$
4: Update($x, y, \mathcal{M}$)
5: $X, Y \leftarrow$ RandomSubset($\mathcal{M}$, n)
6: $g \leftarrow \nabla \ell_\theta(x, y)$; $G \leftarrow \nabla_\theta \ell(X, Y)$
7: $c = \max_i(\frac{\langle g, G_i \rangle}{\|g\|\|G_i\|}) + 1$ ▷ make the score positive
8: **if** len($\mathcal{M}$) $>= M$ **then**
9:     **if** $c < 1$ **then**            ▷ cosine similarity $< 0$
10:         $i \sim P(i) = \mathcal{C}_i / \sum_j \mathcal{C}_j$
11:         $r \sim$ uniform$(0, 1)$
12:         **if** $r < \mathcal{C}_i / (\mathcal{C}_i + c)$ **then**
13:             $\mathcal{M}_i \leftarrow (x, y)$; $\mathcal{C}_i \leftarrow c$
14:         **end if**
15:     **end if**
16: **else**
17:     $\mathcal{M} \leftarrow \mathcal{M} \cup \{(x, y)\}$; $\mathcal{C} \cup \{c\}$
18: **end if**

---

## 4 Experiments

This section serves to validate our approach and show its effectiveness at dealing with continual learning problems where task boundaries are not available.

**Benchmarks**

We consider 3 different benchmarks, detailed below.

**Disjoint MNIST:** MNIST dataset divided into 5 tasks based on the labels with two labels in each task. We use 1k examples per task for training and report results on all test examples.

**Permuted MNIST:** We perform 10 unique permutations on the pixels of the MNIST images. The permutations result in 10 different tasks with same distributions of labels but different distributions of the input images. Following [10], each of the task in permuted MNIST contains only 1k training examples. The test set for this dataset is the union of the MNIST test set with all different permutations.

**Disjoint CIFAR-10:** Similar to disjoint MNIST, the dataset is split into 5 tasks according to the labels, with two labels in each task. As this is harder than mnist, we use a total of 10k training examples with 2k examples per task.

In all experiments, we use a fixed batch size of 10 samples and perform few iterations over a batch (1-5), note that this is different from multiple epochs over the whole data. In disjoint MNIST, we report results using different buffer sizes in table 1. For permuted MNIST results are reported using buffer size 300 while for disjoint CIFAR-10 we couldn't get sensible performance for the studied methods with buffer size smaller than 1k. All results are averaged over 3 different random seeds.

**Models** Following [10], for disjoint and permuted MNIST we use a two-layer neural network with 100 neurons each while for CIFAR-10 we use ResNet18. Note that we employ a shared head in the incremental classification experiments, which is much more challenging than the multi-head used in [10]. In all experiments, we use SGD optimizer with a learning rate of $0.05$ for disjoint MNIST and permuted MNIST and $0.01$ for disjoint Cifar-10.

Table 1: Average test accuracy of sample selection methods on disjoint MNIST with different buffer sizes.

| Buffer Size / Method | 300 | 400 | 500 |
|---|---|---|---|
| `Rand` | $37.5 \pm 1.3$ | $45.9 \pm 4.8$ | $57.9 \pm 4.1$ |
| `GSS-IQP(ours)` | $75.9 \pm 2.5$ | $82.1 \pm 0.6$ | $84.1 \pm 2.4$ |
| `GSS-Clust` | $75.7 \pm 2.2$ | $81.4 \pm 4.4$ | $83.9 \pm 1.6$ |
| `FSS-Clust` | $75.8 \pm 1.7$ | $80.6 \pm 2.7$ | $83.4 \pm 2.6$ |
| `GSS-Greedy(ours)` | $\mathbf{82.6 \pm 2.9}$ | $\mathbf{84.6 \pm 0.9}$ | $\mathbf{84.8 \pm 1.8}$ |

Table 2: Comparison of different selection strategies on permuted MNIST benchmark.

| Method | T1 | T2 | T3 | T4 | T5 | T6 | T7 | T8 | T9 | T10 | Avg |
|---|---|---|---|---|---|---|---|---|---|---|---|
| Rand | $67.01 \pm 2.7$ | $62.18 \pm 4.6$ | $69.63 \pm 3.2$ | $62.05 \pm 2.4$ | $68.41 \pm 1.0$ | $72.81 \pm 3.0$ | $77.67 \pm 2.3$ | $77.28 \pm 1.8$ | $83.92 \pm 0.6$ | $84.52 \pm 0.3$ | $72.54 \pm 0.4$ |
| GSS-IQP (ours) | $74.1 \pm 2.2$ | $69.73 \pm 0.6$ | $70.77 \pm 4.9$ | $70.5 \pm 2.5$ | $73.34 \pm 4.8$ | $78.6 \pm 2.8$ | $81.8 \pm 0.6$ | $81.8 \pm 0.7$ | $86.4 \pm 0.8$ | $85.45 \pm 0.4$ | $77.3 \pm 0.5$ |
| GSS-Clust | $75.3 \pm 1.3$ | $75.22 \pm 1.9$ | $76.66 \pm 0.9$ | $75.09 \pm 1.6$ | $78.76 \pm 0.9$ | $81.14 \pm 1.1$ | $81.32 \pm 2.0$ | $83.87 \pm 0.7$ | $84.52 \pm 1.2$ | $85.52 \pm 0.6$ | $\mathbf{79.74 \pm 0.2}$ |
| FSS-Clust | $82.2 \pm 0.9$ | $71.34 \pm 2.3$ | $76.9 \pm 1.3$ | $70.5 \pm 4.1$ | $70.56 \pm 1.4$ | $74.9 \pm 1.5$ | $77.68 \pm 3.3$ | $79.56 \pm 2.6$ | $82.7 \pm 1.5$ | $85.3 \pm 0.6$ | $77.8 \pm 0.3$ |
| GSS-Greedy (ours) | $83.35 \pm 1.1$ | $70.84 \pm 1.3$ | $72.48 \pm 1.7$ | $70.5 \pm 3.4$ | $72.8 \pm 1.7$ | $73.75 \pm 3.8$ | $79.86 \pm 1.8$ | $80.45 \pm 2.9$ | $82.56 \pm 1.1$ | $84.8 \pm 1.6$ | $77.3 \pm 0.5$ |

Table 3: Comparison of different selection strategies on disjoint CIFAR10 benchmark.

| Method | T1 | T2 | T3 | T4 | T5 | Avg |
|---|---|---|---|---|---|---|
| `Rand` | $0 \pm 0.0$ | $0.49 \pm 0.4$ | $5.68 \pm 4.4$ | $52.18 \pm 0.8$ | $84.96 \pm 4.4$ | $28.6 \pm 1.2$ |
| `GSS-Clust` | $0.35 \pm 0.5$ | $15.27 \pm 8.3$ | $7.96 \pm 6.3$ | $9.97 \pm 2.1$ | $77.83 \pm 0.7$ | $22.5 \pm 0.4$ |
| `FSS-Clust` | $0.2 \pm 0.2$ | $0.8 \pm 0.5$ | $5.4 \pm 0.7$ | $38.12 \pm 5.2$ | $87.90 \pm 3.1$ | $26.7 \pm 1.5$ |
| `GSS-Greedy(ours)` | $42.36 \pm 12.1$ | $14.61 \pm 2.7$ | $13.60 \pm 4.5$ | $19.30 \pm 2.7$ | $77.83 \pm 4.2$ | $\mathbf{33.56 \pm 1.7}$ |

## 4.1 Comparison with Sample Selection Methods

We want to study the buffer population in the context of the online continual learning setting when no task information are present and no assumption on the data generating distribution is made. Since most existing works assume knowledge of task boundaries, we decide to deploy 3 baselines along with our two proposed methods[2] . Given a fixed buffer size $M$ we compare the following:

**Random** (`Rand`): Whenever a new batch is received, it joins the buffer. When the buffer is full, we randomly select samples to keep of size $M$ from the new batch and samples already in buffer.

**Online Clustering**: A possible way to keep diverse samples in the buffer is online clustering with the goal of selecting a set of $M$ centroids. This can be done either in **the feature space** (`FSS-Clust`), where we use as a metric the distance between the samples features, here the last layer before classification, or in **the gradient space** (`GSS-Clust`), where as a metric we consider the Euclidean distance between the normalized gradients. We adapted the doubling algorithm for incremental clustering described in [3].

**IQP Gradients** (`GSS-IQP`): Our surrogate to select samples that minimize the feasible region described in Eq.9 and solved as an integer quadratic programming problem. Due to the cost of computation we report our `GSS-IQP` on permuted MNIST and disjoint MNIST only.

**Gradient greedy selection** (`GSS-Greedy`): Our greedy selection variant detailed in Algo. 2. Note that differently from previous selection strategies, it doesn't require re-processing all the recent and buffer samples to perform the selection which is a huge gain in the online streaming setting.

## 4.2 Performance of Sample Selection Methods

Tables 1, 2, 3 report the test accuracy on each task at the end of the data stream of disjoint MNIST, permuted MNIST, disjoint CIFAR-10 sequentially. First of all, the accuracies reported in the tables might appear lower than state of the art numbers. This is due to the strict online setting, we use shared head and more importantly we use no information of the task boundary. In contrast, all previous works assume availability of task boundary either at training or both training and testing. The performance of the random baseline `Rand` clearly indicates the difficulty of this setting. It can

Table 4: Comparison with reservoir sampling on different imbalanced data sequences from disjoint MNIST.

| Method | Seq1 | Seq2 | Seq3 | Seq4 | Seq5 | Avg |
|---|---|---|---|---|---|---|
| `Reservoir` | 63.7±0.8 | 69.4 ± 0.7 | 66.8±4.8 | 69.1±2.4 | 76.6±1.6 | 69.12±4.3 |
| `GSS-IQP(ours)` | **75.9 ± 3.2** | 76.2± 4.1 | 79.06±0.7 | 76.6±2.0 | 74.7± 1.8 | 76.49±1.4 |
| `GSS-Greedy(ours)` | 71.2±3.6 | **78.5± 2.7** | **81.5±2.3** | **79.5±0.6** | **79.1±0.7** | **77.96± 3.5** |

be seen that both of our selection methods stably outperform the different buffer sizes on different benchmarks. Notably, the gradient based clustering `GSS-Clust` performs comparably and even favorably on permuted MNIST to the feature clustering `FSS-Clust` suggesting the effectiveness of a gradient based metric in the continual learning setting. Surprisingly, `GSS-Greedy` performs on par and even better than the other selection strategies especially on disjoint CIFAR-10 indicating not only a cheap but a strong sample selection strategy. It is worth noting that `Rand` achieves high accuracy on T4 and T5 of the Cifar-10 sequence. In fact, this is an artifact of the random selection strategy where at the end of the sequence, the buffer sampled by `Rand` is composed of very few samples from the first tasks, 24 samples from T1, T2 and T3, but more from the recent T4 (127) & T5 (849). As such, it forgets less the more recent task at the cost of older ones.

## 4.3 Comparison with Reservoir Sampling

Reservoir sampling [17] is a simple replacement strategy to fill the memory buffer when the task boundaries are unknown based on the underlying assumption that the overall data stream is i.i.d distributed. It would work well when each of the tasks has a similar number of examples. However, it could lose the information on the under-represented tasks if some of the tasks have significantly fewer examples than the others. In this paper we study and propose algorithms to sample from an imbalanced stream of data. Our strategy has no assumption on the data stream distribution, hence it could be less affected by imbalanced data, which is often encountered in practice.

We test this scenario on disjoint MNIST. We modify the data stream to settings where one of the tasks has an order of magnitude more samples than the rest, generating 5 different sequences where the the first sequence has 2000 samples of the first task and 200 from each other task, the second sequence has 2000 samples from the second task and 200 from others, and same strategy applies to the rest of the sequences. Table 4 reports the average accuracy at the end of each sequence over 3 runs with 300 samples as buffer size. It can be clearly seen that our selection strategies outperform reservoir sampling especially when first tasks are under-represented with many learning steps afterwords leading to forgetting. Our improvement reaches 15%, while we don't show the individual tasks accuracy here due to space limit, it worth noting that reservoir sampling suffers severely on under-represented tasks resulting in very low accuracy.

Having shown the robustness of our selection strategies both `GSS-IQP` and `GSS-Greedy`, we move now to compare with state of the art replay-based methods that allocate separate buffer per task and only play samples from previous tasks during the learning of others.

## 4.4 Comparison with State-of-the-art Task Aware Methods

Our method ignores any tasks information which places us at a disadvantage because the methods that we compare to utilize the task boundaries as an extra information. In spite of this disadvantage, we show that our method performs similarly on these datasets.

**Compared Methods**

`Single`: is a model trained online on the stream of data without any mechanism to prevent forgetting.

`i.i.d.online`: is the Single baseline trained on an i.i.d. stream of the data.

`i.i.d.offline`: is a model trained offline for multiple epochs with i.i.d. sampled batches. As such `i.i.d.online` trains on the i.i.d. stream for just one pass, while `i.i.d.offline` takes multiple epochs.

`GEM` [10]: stores a fixed amount of random examples per task and uses them to provide constraints when learning new examples.

Figure 4: Comparison with state-of-the-art task aware replay methods GEM. Figures show test accuracy.

(a) Disjoint MNIST  (b) Permuted MNIST  (c) Disjoint CIFAR-10

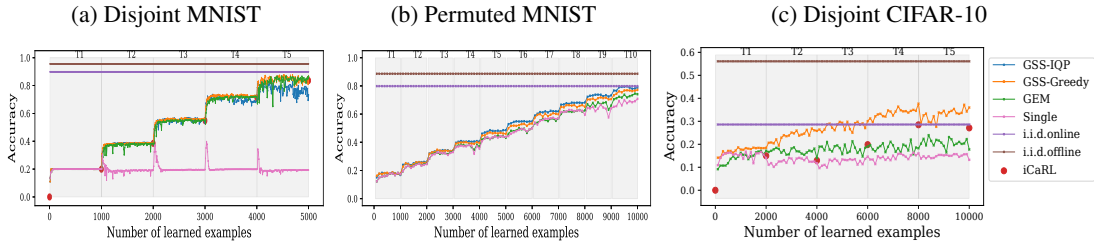

`iCaRL` [14]: follows an incremental classification setting. It also stores a fixed number of examples per class but uses them to rehearse the network when learning new information.

For ours, we report both `GSS-IQP` and `GSS-Greedy` on permuted and disjoint MNIST and only `GSS-Greedy` on disjoint CIFAR-10 due to the computational burden.

Since we perform multiple iterations over a given batch still in the online setting, we treat the number of iterations as a hyper parameter for GEM and `iCaRL`. We found that GEM performance constantly deteriorates with multiple iterations while `iCaRL` improves.

Figure 4a shows the test accuracy on disjoint MNIST which is evaluated during the training procedure at an interval of 100 training examples with 300 buffer size. For the i.i.d. baselines, we only show the achieved performance at the end of the training. For `iCaRL`, we only show the accuracy at the end of each task because `iCaRL` uses the selected exemplars for prediction that only happens at the end of each task. We observe that both variants of our method have a very similar learning curve to GEM except the few last iterations where `GSS-IQP` performance slightly drops.

Figure 4b compares our methods with the baselines and GEM on the permuted MNIST dataset. Note that `iCaRL` is not included, as it is designed only for incremental classification. From the performance of the `Single` baseline it is apparent that permuted MNIST has less interference between the different tasks. Ours perform better than GEM and get close to `i.i.d.online` performance.

Figure 4c shows the accuracy on disjoint CIFAR-10 evaluated during the training procedure at an interval of 100 training examples. `GSS-Greedy` shows better performance than GEM and `iCaRL`, and it even achieves a better average test performance at the end of the sequence than `i.i.d.online`. We found that GEM suffers more forgetting on previous tasks while `iCaRL` shows lower performance on the last task. Note that our setting is much harder than offline tasks training used in `iCaRL` or the multi-heard setting used in .

## 5 Conclusion

In this paper, we prove that in the online continual learning setting we can smartly select a finite number of data to be representative of all previously seen data without knowing task boundaries. We aim for samples diversity in the gradient space and introduce a greedy selection approach that is efficient and constantly outperforming other selection strategies. We still perform as well as algorithms that use the knowledge of task boundaries to select the representative examples. Moreover, our selection strategy gives us advantage under the settings where the task boundaries are blurry or data are imbalanced.

## Acknowledgements

Rahaf Aljundi is funded by FWO.

## Footnotes

[2]The code is available at `https://github.com/rahafaljundi/Gradient-based-Sample-Selection`

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
