[Supplementary Material · SampleSelection_Supp.pdf]

# Gradient based sample selection for online continual learning - Supplementary Materials

## 1 Clarifications of points in the main paper

### 1.1 Estimation of the solid angle

We faced the problem of no tractable formula for the solid angle, so we estimated it with sampling method. The estimated angle is an average of Bernoulli random variables with variance then bounded by $\frac{1}{4N}$ where $N$ is the number of these Bernoulli variables. By taking $N = 10^9$, we reach an asymptotic confidence interval of length around $10^{-4}$.

### 1.2 IQP formulation to our surrogate

Our surrogate formulation for selecting a fixed set of samples that minimize the solid angle is:

$$\text{minimize}_{\mathcal{M}} \sum_{i,j \in \mathcal{M}} \frac{\langle g_i, g_j \rangle}{\|g_i\|\|g_j\|} \tag{1}$$
$$s.t. \ \mathcal{M} \subset [0 \mathinner{.\,.} t-1]; \ |\mathcal{M}| = M$$

We solve the surrogate minimization as an integer quadratic programming problem. We first normalize the gradients: $G = \frac{\langle g_i, g_j \rangle}{\|g_i\|\|g_j\|}$ and find a selection vector $X$ that minimizes the following:

$$\underset{X}{\text{minimize}} \quad \frac{1}{2} X^T G X$$
$$s.t. \quad \mathbf{1}^T . X = M$$
$$x_i \in \{0, 1\} \quad \forall x_i \in X$$

where $\mathbf{1}$ is a vector of ones with the same length as $X$. Selected samples will correspond to values of 1 in $X$.

## 2 Additional Experiments

### 2.1 Performance under blurry task boundary

An interesting setting is the scenario where there are no clear task boundaries in the data stream, as we mentioned in the introduction, such situation can happen in practice. We start by blurring the task boundaries in disjoint Cifar10 benchmark. For each task in a dataset, we keep the majority of the examples while we randomly swap a small percentage of the examples with other tasks. A larger swap percentage corresponds to more blurry boundaries. A similar setting has been used in [1]. We keep 90% of the data for each task, and introduce 10% of data from the other tasks. We make comparisons to the other studied selection methods. Since tasks are not disjoint, forgetting is not as

Figure 1: Greedy Sample Selection Ablation Study. Figures show test accuracy.

(a) Average test accuracy at the end of disjoint MNIST for different values of $n$.

(b) Average test accuracy at the end of disjoint MNIST for different batch sizes.

sever as complete disjoint tasks. Hence, we use a buffer of 500 samples and train on 1k samples per task which allows us to run our `GSS-IQP` more smoothly.

Table 1 reports the accuracy of each task at the end of the sequence. Our both methods perform better than other selection strategies.

| Method | T1 | T2 | T3 | T4 | T5 | Avg |
|---|---|---|---|---|---|---|
| Rand | 0 | 3.45 | 9.85 | **54.67** | 78.76 | 29.0 |
| GSS-IQP(ours) | 9.38 | **11.33** | **17.05** | 30.84 | **79.53** | **29.6** |
| GSS-Clust | 2.43 | 16.75 | 9.09 | 20.71 | 77.98 | 25.0 |
| FSS-Clust | 2.95 | 05.09 | 6.06 | 38.16 | 78.14 | 26.0 |
| GSS-Greedy(ours) | **34.2** | 11.14 | 14.96 | 20.25 | 67.5 | **29.6** |

Table 1: Comparison of different selection strategies on disjoint Cifar10 with blurry task boundary.

## 2.2 Constrained Optimization Compared to Rehearsal

By the end of the section **??**, we have elaborated on the computational complexity of the constrained optimization with large buffers which renders infeasible. That's mainly because at each learning step, gradients of each sample in the buffer needs to be estimated and then the new sample gradient need to be projected onto the feasible region determined by all the samples gradients. As an alternative, we perform rehearsal on the buffer. Here, we want to compare the performance of the two update strategies, the constrained optimization `GSS-IQP(Constrained)` and the rehearsal `GSS-IQP(Rehearsal)`. We consider disjoint MNIST benchmark and use 200 training samples per task. Table 2 reports the test accuracy on each task achieved by each strategy at the end of the training when using a buffer of size 100 while table 3 reports the accuracy with 200 buffer size. `GSS-IQP(Constrained)` improves over `GSS-IQP(Rehearsal)` with a margin of $3 - 5\%$ but requires a long time to train as it scales polynomialy with the number of samples in the buffer apart from the need to compute the gradient of each buffer sample at each training step. `GSS-IQP(Rehearsal)` with larger buffer is less computational and yields similar results, comparing `GSS-IQP(Rehearsal)` with a buffer of size 200 $(78.9\%)$ and `GSS-IQP(Constrained)` with a buffer of size 100 $(76.26\%)$.

## 2.3 Effect of $n$ in Greedy Sample Selection

In our Greedy Sample Selection (`GSS-Greedy`), for each newly received sample we compute a score indicating its similarity to samples already in the buffer (lines (5-7) in Algorithm **??**). This is done by computing the cosine similarity of the new sample(s) gradient to $n$ gradient vectors of samples drawn from the buffer. We study the effect of $n$ on the performance of GSS-Greedy given Disjoint MNIST benchmark.

Figure 1a shows the average test accuracy at the end of disjoint MNIST sequence for different values of $n$. Very small value of $n$ tends to give a very noisy estimate of the new sample(s) similarity score

| Method | T1 | T2 | T3 | T4 | T5 | Avg |
|---|---|---|---|---|---|---|
| GSS-IQP(Constrained) | 90.0 | 70.0 | 45.13 | 88.77 | 86.08 | 76.26 |
| GSS-IQP(Rehearsal) | 81.5 | 69.47 | 46.96 | 69.80 | 88.0 | 71.3 |

Table 2: Comparison between our GSS-IQP constrained and GSS-IQP rehearsal, buffer size 100.

| Method | T1 | T2 | T3 | T4 | T5 | Avg |
|---|---|---|---|---|---|---|
| GSS-IQP(Constrained) | 95.0 | 83.0 | 68.7 | 87.6 | 82.4 | 83.4 |
| GSS-IQP(Rehearsal) | 94.6 | 83.89 | 50.6 | 77.0 | 88.67 | 78.9 |

Table 3: Comparison between our GSS-IQP constrained and GSS-IQP rehearsal, buffer size 200.

and hence new samples are added more often to the buffer resulting in a more forgetting of previous tasks and a less average test accuracy, $Avg.Acc. = 67.3$ for $n = 1$. Increasing $n$ tends to give a better estimation and as a result better average test accuarcies. However, large values of $n$ lead to a high rejection rates as it only adds new samples that are very different from all samples in buffer. As such, first tasks will have more representative samples in the buffer than later tasks and average test accuracy slightly decreases. In all the experiments, we used $n = 10$ as a good trade-off between good score approximation and computational cost.

## 2.4 Batch size effect.

In this paper, we consider a never-ending stream of data that we aim at learning efficiently. In our experiments, we wanted to be as close as possible to the full online setting. Hence, we used a batch size of 10, as in GEM [2], which seems a good approximation. To test the effect of the batch size on the our continual learning performance, we run GSS-Greedy on disjoint MNIST benchmark with buffer size $M = 300$ considering different batch sizes.

Figure 1b shows the average test accuracy at the end of the sequence for different batch sizes. In the online learning, only one pass over the training data of a given "task" is performed. Large batch sizes lead less parameter updates. As a result, the learning methods fails to achieve a good performance on the learned samples compared to the configuration with smaller batch size. Additionally, very small batch size means noisier parameter update estimation.