[Reviews · NeurIPS 2019]

Reviewer 1



This paper proposes an approach to optimally select samples for a small replay buffer to perform continual learning (CL) without forgetting. Like previous work (eg. GEM/A-GEM) the problem is formulated from the perspective of constrained optimisation (minimise loss on current sample subject to loss not increasing on previous ones). Unlike GEM, with clear separation and knowledge of tasks, this approach addresses the general non-stationary learning problem. The paper proposes a theoretical argument for using the variance of gradients to select samples for the buffer. One related work that could also be cited is "Adapting Auxiliary Losses using Gradient Similarity", by Du et al, 2018 (https://arxiv.org/abs/1812.02224) I think this is a novel idea that is well worth exploring and could be a nice publication, but have a number of questions and concerns that I believe need to be addressed first. On the technical side: - What if no feasible region exists? Eg. for a binary classification problem, two [almost] identical samples with label noise would be enough for the feasible region C to be the empty set (as the gradients would be exactly opposed). - One concern I have is with the successive approximations made. We want Ctilde to match C as closely as possible, this is then relaxed to Ctilde being as small as possible --> minimising solid angle --> minimising the surrogate objective. Fig 2 shows the accuracy of the last step (and in fact, how well does this work for a bigger range of angle/surrogate values?); but how does this translate to approximation error in the original objective? Which constraints are typically satisfied or violated (ie. are there qualitative differences in the kinds of samples that are remembered and forgotten)? - Given an impetus to minimise "size" how do we stop Ctilde choosing a [close to] empty set (eg. Two opposing constraints from noisy data as suggested above) - The iCARL CIFAR-10 results reported here seems poorer than even the CIFAR-100 result (ie. a much harder task) reported in the original paper - where does this discrepancy arise? - Can you quantify the computational improvement of GSS-greedy over the others? This is just stated in passing in the text in lines 220-222. - A number of the reproducibility checklist options haven't been satisfied - things like error bars / uncertainty, hardware used for experiments, specification of evaluation protocol, etc. Most of these are easy, and I think uncertainty over multiple runs is important (at the very least for external comparisons) - Additional benchmarks would be nice in order to contextualize these results - currently it compares against GEM and iCARL. On the writing side: - The related work section is extremely brief, and the paper needs to be better positioned, first within CL, and then constraint-based CL. - The intro seems to be quite critical of prior-focused methods and needs more citations, and possibly toned down language (some of it seems emotive, such as "enjoy the beauty of...", "Prior-focused method has bad performances..." - Figure 1 is not immediately clear, I would show the blue constraint on just the left hand plot to indicate it is being removed on the right. (Showing on a plot something that is removed is a bit difficult to follow) - I'm still not clear on what iid online (as opposed to offline) refers to. Is it just training a single model on iid data for the same number of steps as the incremental setting? If so, why not start the curves at zero as well for consistency? - While the intro motivates the approach broadly as relaxing i.i.d assumptions within each task (lines 46-50), the only experiments performed are with a sequences of tasks, each shown i.i.d. As such, I think the motivation / claims need to be narrower, or additional non-stationary scenarios evaluated. Other minor points: - Avoid conversational or imprecise language, eg. "Way smaller" on line 118, and "this exact" in 129. - There are also some minor grammar issues throughout (such as "bad performances" in line 52), so please double check for consistency. - Line 151: "as shown in Suppl" doesn't make sense. - Is "disjoint MNIST" not the same as Split MNIST? If so, I'd change it to be consistent POST-REBUTTAL: After looking at all of the reviews and the rebuttal, I don't think my position has really changed. Our concerns seem to be mainly that it would be nice to have more experiments on some other standard benchmarks, as well as some additional discussion and analysis on the high-level picture, particularly in terms of how the different steps and approximations behave and interact. I feel this is a nice idea, but there's more the authors could do to make it even more convincing. As such, I'm thinking of staying with my original score of 6.

Reviewer 2



**** I thank the authors for the response. My concerns have been partly addressed and thus I will increase my score. However, the submission would benefit from a round of clean-up: clearly state how each approximation step works and how well this works, the relationship to existing methods, and add some existing benchmarks (if possible) as this is key for community adoption/acceptance. **** Originality: The formulation of the sample selection as a feasible region selection seems sensible. The proposed IQP and greedy algorithms seem sound. However, it is unclear how well these actually work theoretically in high dimensional space and when the buffer size is large. The interpretation of the proposed surrogate function as maximising the diversity of the samples in the replay buffer is interesting but begs the question that there is some similarity between the proposed methods and existing coreset selection algorithms, for example: the K-center algorithm in "Teofilo F. Gonzalez. Clustering to minimize the maximum intercluster distance", or more recent coresets algorithms in "Bachem et al. Practical coreset constructions for machine learning". Quality: It is challenging to judge whether the proposed method actually performs well in a more realistic continual learning setting, due to the difference between the experiments considers and existing benchmarks. The number of random runs considered is also quite small and that the result tables do not include error bars (it would be better to convert these tables into figures, as it is much easier in general to understand the real difference between methods -- there are more than just one single bold number). It is not clear how well the empirical surrogate work in high dimensions, or that how accurate the proposed greedy strategy is. It would also be interesting to have a more in-depth discussion/comparison to existing methods such as coverage maximisation as briefly mentioned on line 72. The paper seems to criticize methods that combine episodic memory/experience replay with parameter based regularisation (in the intro) -- but isn't the final algorithm considered in this paper a form of such algorithm? [the line is pretty blurred, but the buffer can be used for objective/gradient regularisation or replay as discussed in 3.6]. In general, I also feel there is not enough high level picture on how the algorithms are derived and to glue different subsections together, which makes it hard to read. The claim in the conclusion "efficient and constantly outperforming other selection strategies" seems very strong given the mixed performance of both IQP and greedy in many tasks considered. Clarify: The writing is clear, in general. However, there are presentation styles in the experiments that make things quite hard to read to me. For example, the tables are not in order, the subheadings in 4 and 4.4 are hard to read/look strange. In figure 2, what is "200D log scale". Line 129: "this exact in 2D with figure 3"?.

Reviewer 3



Originality: This paper proposes a new method for sample selection in continual learning, which is based on GEM and reformulates the problem from the perspective of constrained optimization. Comparing with other related works, the experiments in this paper have been conducted in a different setting of continual learning, that is, the task boundary is unavailable and the model can only iterate on a mini-batch for a few times. This setting is worth to bring out as it is more likely in the real world. Quality: The paper is well written and easy to follow. In my understanding, the intuition is straightforward and the formulation is reasonable. In the experiments, the proposed methods have been compared with several different strategies of sample selection as well as some existing methods (i.e. GEM and iCaRL) by average accuracy. It could be more complete by comparing Backward Transfer with GEM since GEM has shown an advantage in it and this work is based on GEM. Clarity: The problem setting and formulations are clear. However, there are still some issues I would like the authors to clarify a bit: 1. Is it possible to give a consistency analysis for the greedy method (Alg. 2)? As increasing n to M, is it equivalent to the IQP method? 2. What's the value of n you used in the greedy method in your experiments? Is this value critical to the performance of the greedy method? 3. You used a batch size of 10 samples in all experiments, which is quite small, did you try other sizes? Would a larger size perform better or not? 4. The results were averaged over 3 runs, but how about the standard deviation? Does the greedy method have a larger variance than other methods? 5. In Tab. 3 T4, the random selection has much higher accuracy than others, do you have any insights on this? Significance: The proposed methods show promising results compared with some state-of-the-art approaches and the greedy one shows practical feasibility for large scale buffers. The paradigm of continual learning used in this paper could be preferred in some real-world applications. ############################# I generally feel no reason to change my score after reading authors response. The extra experiment results are quite interesting, the performance decreasing while increasing n. It would be great if authors can provide some insights on it.

[Author Response · NeurIPS 2019]

Table 1: $\mu$ accuracies $\pm\sigma$ of the main experiments.

|  | dis. MNIST, M=300 | perm. MNIST | cifar-10 |
|---|---|---|---|
| Random | $37.5 \pm 1.35$ | $72.7 \pm 1.45$ | $28.6 \pm 1.23$ |
| GSS-IQP | $75.9 \pm 2.54$ | $77.3 \pm 0.54$ | - |
| GSS-Clust | $75.7 \pm 2.17$ | $79.9 \pm 0.69$ | $22.5 \pm 0.43$ |
| FSS-Clust | $75.8 \pm 1.56$ | $77.8 \pm 0.33$ | $26.7 \pm 1.47$ |
| GSS-Greedy | $82.6 \pm 2.9$ | $77.3 \pm 0.50$ | $33.5 \pm 1.62$ |

We thank the reviewers for their valuable comments. We want
to emphasize that our work is the first to tackle the problem
of online continual learning with *no task boundaries*, i.e. dif-
ferently from other methods e.g. GEM, iCaRL, we receive no
information when task T1 switches to T2, which makes the
input stream highly non iid at the boundary. Our method is thus applicable to scenarios of continual learning when task
id is unknown. We will expand the related work and improve the presentation of the figures and the tables.
**Error bars? larger variance for greedy?** Tab. 1 reports the main results with their standard deviation estimated over
the different runs. This confirms that the advantage of our method is statistically significant.
R1: **What if no feasible region exists (empty $\tilde{C}$)?** This is a good point. When the data contains outliers it is an
open question how to distinguish outliers from under-represented examples. As for this writing, we don't consider the
existence of outliers. Nevertheless, our method should still work with outliers: 1) Our surrogate considers pairs of
samples, although the outlier is likely to be selected, pairs that don't contain the outliers will not be affected. 2) Rigorous
satisfaction of the constraints is not possible in non-convex neural networks, it is converted to soft regularization.
**Fig 2 , bigger range of angle/surrogate values?** The range of the solid angle is already from 0 to 1 which is the full
range of solid angle. In the log scale it may seem truncated because we omit zeros which is negative infinity in log scale.
**Which constraints are typically satisfied or violated?** The samples are selected to be diverse, and the discarded
samples usually have similar counterparts in the buffer. We found the selected samples to cover the different patterns
learned classes leading to a balanced buffer even with an imbalanced data stream.
**iCaRL CIFAR-10 low.** We consider online incremental classification, differently from iCaRL, Ln 294. iCaRL performs
offline training with large batches and repeatedly revisiting over the classes of a task.
**Computational improvement of GSS-greedy?** For our greedy alternative, the major cost corresponds to the estima-
tion of the n gradients of the drawn samples, n was fixed to 10 in all our experiments. For clustering based sample
selection, the features/gradients of the buffer samples and the corresponding distances need to be computed at each step.
R2: **Clarification on high level picture** We start from the constrained optimization view of continual learning like in
GEM, which needs to be relaxed by constraint selection. Instead of random selection, we perform constraint selection
by minimizing the solid angle formed by the constraints. We propose a surrogate for the solid angle objective, and show
their relation numerically. Finally we test the effectiveness of the surrogate on continual learning benchmarks.
**High dimensional space and when the buffer is large?** We only perform the numerical analysis of the monotonous
correlation of the solid angle and the surrogate up to 200 dimension, because the complexity of Monte-Carlo integration
increases exponentially with the dimension. Empirically we verified with experiments with a buffer size up to 1k and
show it is still effective. We also showed another interpretation of our surrogate i.e. to maximize the diversity of the
samples in the "gradient" space. We believe diversity maximization is likely to still work in high dimensional space.
**Maximising diversity is similar to existing coreset algorithms / coverage maximisation?** A key difference is that
we consider the diversity in the gradient space instead of data space. The coverage maximization reported are performed
in the data space using euclidean metric, which is not working equally well for high dimensional data like image, (Fig. 7
in the coverage maximisation paper). Our contribution is the proposal of using gradient information to measure diversity
which is supported by the constrained optimization (Eq. 1). We considered two incremental clustering baselines, in the
feature space and in the gradient space, and showed that our approach is superior (Tab. 1,3,4 in the paper).
**Methods that combine replay with parameter regularisation criticized?** We state that hybridizing prior focused
with replay based is necessary to achieve better performance on long sequences, Ln 27-29. Also, there's a difference
between the two "regularizations". The one mentioned in the intro is the prior focused regularizing the parameters while
the regularization mentioned later is about converting the hard constraint into a regularization term i.e. rehearsal.
**More standard benchmarks?** We opt for more realistic setting with online training with non stationary distribution
and never ending training. This is already the standard setting considered in GEM but with multi-head. Our method
aims to solve the hard setting and not the task incremental setting where a task oracle is used at training and test time.
R3: **Consistency analysis for the greedy method? As increasing n to M, is it equivalent to the IQP method?** For
greedy, the drawn samples are used to estimate a score of the new sample indicating its similarity to the buffer samples.
IQP is exact and solves the surrogate, Eq.7 at each step. Greedy keeps scores of buffer samples once added which relate
to their chance of being replaced. We fixed n to 10 on all settings. We abate the effect of n with dis. MNIST, M= 300.
We get the following accuracies n=1:67.3, n=5:79.0, n=10:82.6, n=20:79.6, n=30:79.9, n=40:78.4, n=50:78.3.
**Batch size effect.** We want to be as close as possible to the full online setting. Batch size of 10, used in GEM, seems a
good approximation. We run dis.MNIST, M=300, with different batch sizes. The mean accuracies of GSS-greedy are
B=5:82.9, B=10:82.6, B=20:84.23, B=50: 80.2, B=100:72.9. Large batch sizes lead less parameter updates.
**Random high accuracy on T4.** Its buffer has very few samples from previous tasks, 24 for T1-T3, and more from the
recent T4 (127) & T5 (849) given M= 1k. As such, it forgets less the more recent task at the cost of older ones.
**Backward Transfer.** It is shown in GEM with multi head, i.e. when evaluating a task, only its classes are considered.
Under shared head this is different. When a task is first evaluated, fewer classes are learned and others will have low
accuracy. At the end of sequence, all classes compete which makes backward transfer hard to achieve.

[Meta-Review · NeurIPS 2019]

There is an interesting contribution in terms of formulating the continual learning setting into a geometry-based selection problem of samples from the experience replay. Reviewers would like to see more experiments on other standard benchmarks, as well as additional discussion about how each approximation step works and the relationship to other existing methods.